# BCAA (Branched-Chain Amino Acids) Inhibiting the Autophagy System via the Activation of mTORC1, Thereby Upregulating the Tumor Suppressor PDCD4 in Huh7 Hepatoma Cells

**DOI:** 10.3390/cells14241975

**Published:** 2025-12-11

**Authors:** Rasheda Perveen, Iwata Ozaki, Hirokazu Takahashi, Md Manirujjaman, Takuya Kuwashiro, Sachiko Matsuhashi

**Affiliations:** Division of Hepatology, Diabetology, Endocrinology and Metabolism, Department of Internal Medicine, Faculty of Medicine, Saga University, 5-1-1 Nabeshima, Saga 849-8501, Japan; ozaki@cc.saga-u.ac.jp (I.O.); mmanir@lsuhsc.edu (M.M.); f8451@cc.saga-u.ac.jp (T.K.); matsuha2@cc.saga-u.ac.jp (S.M.)

**Keywords:** BCAA, PDCD4, autophagy, mTORC1, Huh7

## Abstract

Branched-chain amino acids (BCAAs) are essential amino acids in humans, with reported anti-proliferative effects on HepG2 hepatoma cells and the potential to reduce hepatocellular carcinoma (HCC) development in cirrhotic patients. PDCD4, a tumor suppressor that is downregulated in many cancers, is also suppressed by serum, EGF, or TPA treatment. This study examined the effect BCAA has on PDCD4 expression and related cellular pathways in Huh7 hepatoma cells. Cells were treated with different concentrations of BCAA, and analyzed by Western blotting, qRT-PCR, and immunofluorescence staining. Treatment with BCAA upregulated the protein levels of PDCD4, while downregulating its mRNA levels. BCAA enhanced the phosphorylation of mTORC1 substrate 4E-BP1, p70S6K1, and p70S6K1 substrate S6 ribosomal protein. BCAA also elevated the protein levels of autophagy factors p62 and ATG5 while reducing LC3-II particle formation, thus indicating impaired autophagy. ULK1 knockdown also upregulated the protein levels of PDCD4 and p62. Additionally, BCAA upregulated the phosphorylation of ULK1 at serine 757, which was inhibited by rapamycin. These findings suggest that BCAA inhibits autophagy through the mTORC1-mediated phosphorylation of ULK1 at serine 757, thereby impairing autophagosome formation and upregulating the PDCD4 protein levels by inhibiting its degradation via autophagy. Furthermore, FACS analysis showed that BCAA inhibited the proliferation of Huh7 cells. BCAA may have a preventive effect against tumor development through the modulation of autophagy and the tumor suppressor pathways.

## 1. Introduction

Branched-chain amino acids (BCAAs), leucine (Leu), isoleucine (Ile), and valine (Val) are essential amino acids for humans and control the metabolism of proteins, glucose, and fats [1,2,3,4,5]. BCAA supplementation and BCAA-rich diets have been shown to be useful for regulating body weight, muscle protein synthesis, and glucose homeostasis [3,6,7,8]. In vitro studies have shown that BCAA inhibits the proliferation of HepG2 hepatoma cells [9]. In addition, BCAA supplementation prevents the development of liver tumorigenesis and improves insulin resistance in obese and diabetic mice by decreasing the expression of insulin-like growth factor (IGF)-1, IGF-2, and IGF-1 receptor [10]. Clinical trials have reported that BCAA supplementation reduces hepatocellular carcinoma (HCC) development in patients with cirrhosis [1,11]. These data indicate that BCAA supplementation may be useful as a preventive therapy. Conversely, the circulating BCAA levels tend to be elevated in obese individuals, and their elevated circulating levels are closely related to future insulin resistance (IR) or type 2 diabetes mellitus (T2DM) [12,13]. In genetically obese (ob/ob) mice, rate-limiting branched-chain α-keto acid (BCKA) dehydrogenase deficiency accumulates BCAA and BCKA. Due to corrections of BCAA catabolic defects, an elevation of the BCAA/BCKA levels both abolishes and attenuates the insulin resistance (IR) of the mice. Similar outcomes with improved insulin sensitivity were obtained by reducing the protein (and thus BCAA) intake in the mice [14].

BCAA activates the mechanistic target of the rapamycin complex 1 (mTORC1) signaling pathway [15,16]. It was shown that, in the BCAA catabolism lost in patients with tumors, mTORC1 activity is enhanced and contributes to an increase in the tumor aggressiveness and disease progression [15,17]. The nutrient-dependent translocation of mTORC1 to the lysosomal surface is a crucial step for the activation of mTORC1 [18]. Upstream of mTOR, amino acids have been shown to activate mTORC1 through an amino acid-sensing cascade involving RAG GTPases [18,19,20], which are regulated by several factors, including GATOR 1, 2, 3 and CASTOR [21,22]. Sestrin2 was identified as the leucine (a member of BCAA) sensor [23]. Sestrin2 is associated with GATOR2, and the complex protein inhibits mTORC1 signaling. When leucine is added to the culture medium of cells, leucine binds to sestrin2 and disrupts sestrin2-GATOR2 association, resulting in the activation of mTORC1 [23]. Activated mTORC1 phosphorylates 4E-binding protein-1 (4E-BP1) and activates p70 ribosomal protein S6 kinase 1 (p70S6K1), which phosphorylates S6 ribosomal protein, resulting in the stimulation of protein synthesis [24]. Furthermore, it was shown that activated p70S6K1 phosphorylates insulin receptor substrate (IRS)-1 and (IRS)-2 and the phosphorylated IRSs were degraded in the proteasome system, thus inhibiting the insulin-induced signaling pathway [25,26,27,28,29]. Therefore, the persistent activation of mTORC1 may promote insulin resistance or type 2 diabetes mellitus (T2DM) [27,28,29].

PDCD4 is a novel tumor suppressor with multiple functions that modulate transcription and translation [25]. PDCD4 inhibits cap-dependent translation by inhibiting the RNA helicase activity of eukaryotic translation initiation factor 4A (eIF4A) in the translation initiation factor complex eukaryotic translation initiation factor 4F (eIF4F) [30,31]. PDCD4 expression has been reported to be downregulated in many tumor tissues and a loss of expression correlates with tumor development and progression [32,33,34,35,36,37,38,39]. PDCD4 knockdown mice mostly developed B-cell lymphoma and showed a shorter life span than normal siblings. However, PDCD4-deficient mice are resistant to inflammatory diseases such as autoimmune encephalomyelitis and diabetes [40].

The PDCD4 protein levels are downregulated by treatment with serum [41], the tumor promotors epidermal growth factor (EGF), or 12-O-tetradecanoylphorbol-13-acetate (TPA) [42]. On the phosphorylation at S71 and S76 in the degron sequence, the PDCD4 protein is ubiquitinated by Skp1-Cullin1-F-box protein (SCF) β-transducin repeat-containing protein (SCF^βTRCP^) and degraded in the proteasome system [41]. The ubiquitinated PDCD4 protein is navigated into autophagosomes interacting with p62/SQSTM1 (sequestosome 1) and degraded also in the autophagy system [43]. Because BCAA inhibits the growth of HepG2 cells [9] and PDCD4 is a tumor suppressor, it was expected that PDCD4 may be involved in the BCAA signaling. When investigating the BCAA effect on the Huh7 hepatoma cell growth, BCAA was found for the first time to upregulate PDCD4 protein levels in the cells. We herein investigated the mechanism underlying BCAA-induced PDCD4 upregulation. Furthermore, this study aimed to assess the impact of BCAA on the proliferation of Huh7 hepatoma cells, thereby elucidating the relationship between BCAA-induced PDCD4 regulation and cellular behavior.

## 2. Materials and Methods

### 2.1. Cells

Huh7 cells were purchased from the Japanese Cancer Research Resources Bank (Osaka, Japan). ATG5 mutant-16 Huh7 cells were generated in the same way as stated in [44]. Dulbecco’s Modified Eagle’s Medium (DMEM; Sigma-Aldrich, St. Louis, MO, USA) containing 10% fetal bovine serum (FBS) was used as a growing and maintenance medium for cells in the atmosphere of 5% CO_2_ at 37 °C. RPMI-1640 medium was purchased from Fujifilm (Osaka, Japan).

### 2.2. Antibodies

We bought antibodies from Cell Signaling Technology (Danvers, MA, USA) to test for anti-phospho-p70 S6K1 (T389), anti-p70 S6K1, anti-phospho-Akt (Thr505, Ser643), anti-Akt, anti-phospho-4EBP1 (T37/46), anti-4EBP1 (T46), anti-phospho-S6 (S235/236), anti-S6, anti-ULK1, anti-phospho-ULK1 at Serine 757(pULK1-s757), and anti-β-actin. We obtained mouse anti-LC3, rabbit anti-LC3, and anti-p62 antibodies from MBL (Tokyo, Japan). The Santa Cruz Biotechnology (Dallas, TX, USA) mouse anti-PDCD4 antibody was used. The rabbit anti-PDCD4 antibody was prepared as described previously [38]. PE-conjugated anti-human Ki67 antibody was purchased from BioLegend (San Diego, CA, USA).

### 2.3. Reagents

Bafilomycin A1 (an autophagy inhibitor) was purchased from Sigma-Aldrich. Rapamycin (an mTORC1 inhibitor) and MG132 (a proteasomal inhibitor) were purchased from Calbiochem (San Diego, CA, USA). The Zombie Aqua Fixable Viability Kit (423101) from BioLegend (San Diego, CA, USA) was used to detect the live cells. We obtained the eBioscience FOXp3/Transcription factor staining buffer from Invitrogen (Thermo Fisher Scientific, Waltham, MA, USA). For nuclear staining, 7-amino-actinomycin D (7-AAD) was obtained from Sony Biotechnology (San Jose, CA, USA).

### 2.4. Treatment of Cells with BCAA

We used two media systems for BCAA treatment: RPMI and DMEM. For the RPMI system, Huh7 cells were grown in DMEM with 10% FBS at a density of 1–1.5 × 10^5^ cells/2 mL (35 mm dish) for 3–5 days. They were then washed twice with RPMI-1640 medium and incubated with RPMI-1640 medium for 1 h at 37 °C. After that, the cells were incubated with media containing BCAA as LIVACT compositions (leucine/isoleucine/valine, 2:1:1.2; EA-Pharma Co. Ltd., Tokyo, Japan) [44] or each member (Leu, Ile, and Val) of BCAA. The basic concentrations of each member (×1) were 50 µg/mL Leu, 25 µg/mL Ile, and 30 µg/mL Val. BCAA-containing RPMI medium corresponded to ×3 (150 µg/mL Leu, 75 µg/mL Ile and 90 µg/mL Val) and ×5 (250 µg/mL Leu, 125 µg/mL Ile, and 150 µg/mL Val). RPMI 1640 (50 µg/mL Leu, 50 µg/mL Ile, and 20 µg/mL Val) was used as a control (R). DMEM without BCAA was prepared as follows: DMEM without amino acids (Fujifilm)was supplemented with a DMEM-based amino acid without BCAA (Leu, Ile, and Val) kindly supplied by EA-Pharma Co., LTD, and then BCAA-containing DMEM medium was used; ×0 (no BCAA) was used as a control, ×2 (100 µg/mL Leu, 50 µg/mL Ile, 60 µg/mL Val), and ×4 (200 µg/mL Leu, 100 µg/mL Ile, 120 µg/mL Val).

### 2.5. siRNA-Mediated Knockdown of ULK1

Huh7 cells were transfected with ULK1 siRNA using Lipofectamine RNAiMAX (Invitrogen, Waltham, MA, USA) following the manufacturer’s instructions. Briefly, 1.5–2 × 10^5^ cells were put in 35 mm dishes and cultured for 3–5 days in DMEM complete medium. When the cells reached 80–90% confluency, the media was switched to 10% FBS-containing DMEM without antibiotics. Forward transfection methods were used to transfect cells with a mixture of siRNA-Lipofectamine RNAiMAX (20 nm siRNA/dish). After 24 h, siRNA-transfected cells were collected for Western blotting and qRT-PCR analysis. ULK1-specific siRNA sequences (1) 5′-GUGGCCCUGUACGACUUCCAGGAAAtt-3′ (#1), (2) 5′-GCACAGAGACCGUGGGCAAtt-3′ (#2) [45], and (3) 5′-CGCGGUACCUCCAGAGCAAtt-3′ [46] were obtained from Hokkaido Science System Co. Ltd. (Hokkaido, Japan). Qiagen (Hilden, Germany)-Allstar negative control siRNA (1027281) was used.

### 2.6. Western Blotting

The treated cells, as previously discussed, were harvested and lysed using a sodium dodecyl sulfate (SDS) buffer composed of 50 mM Tris (pH 6.8), 2.3% SDS, and 1 mM phenylmethanesulfonylfluoride (PMSF, a protease inhibitor) supplemented with 1 × phosphatase inhibitor (Thermo Fisher Scientific, Waltham, MA, USA). Centrifugation at 12,000× *g* for 10 min eliminated cell debris, and the supernatant was obtained. The protein concentration was quantified by the Lowry method using the DC^TM^ protein assay kit (Bio-Rad, Hercules, CA, USA). The bovine serum albumin was used as the standard. To obtain an equal volume of sample, about 10–30 µg of protein from each sample was combined with SDS sample buffer, subjected to SDS-polyacrylamide gel electrophoresis, and transferred to a polyvinylidene difluoride (PVDF) membrane (Bio-Rad). The membranes were blocked overnight at 4 °C with a solution of 10% skim milk and 0.1% Tween 20 in phosphate-buffered saline (PBST). Then they were incubated with the primary antibody for 1 h at room temperature or overnight at 4 °C. After washing three times with 0.1% Tween 20 in PBS (PBST), the membranes were stained with horseradish peroxidase-conjugated secondary antibodies for 1 h at room temperature. All immunoblots were visualized using Super Signal™ West Pico PLUS Chemiluminescent Substrate (Thermo Fisher Scientific, Waltham, MA, USA) according to the manufacturer’s instructions. Tris-buffered saline (TBST) was used instead of PBST to detect the phospho-proteins. We employed the anti-β-actin antibody as an endogenous control. The Fuji Medical X-ray film (Tokyo, Japan) was utilized to obtain the bands of stained membrane and the ImageJ software (version 1.54) program [ https://imagej.nih.gov/ij/ (accessed on 30 March 2023)] was used to analyze the individual protein bands on the film and normalized to β-actin.

### 2.7. Quantitative Real-Time Reverse Transcription Polymerase Chain Reaction (qRT-PCR)

For qRT-PCR, total RNA was extracted from treated cells utilizing RNAiso Plus (Takara, Kusatsu, Japan) and then reverse-transcribed to cDNA with a High Capacity cDNA Reverse Transcription Kit (Thermo Fisher Scientific, Waltham, MA, USA) in accordance with the manufacturer’s guideline. We used a QuantStudio 6 Pro (Applied Biosystems, Waltham, MA, USA) for real-time PCR using PowerUp SYBR Green Master Mix (Thermo Fisher, Waltham, MA, USA) as directed by the manufacturer. Primers for PDCD4, ULK1 [47], and GAPDH (OriGene qStar-NM-002046) were produced by Hokkaido System Science Co., Ltd. (Sapporo, Hokkaido, Japan). The primer sequences were as follows: PDCD4, forward (F) 5′-ATGAGCAGATACTGAATGTAAAC-3′ and reverse (R) 5′-CTTTACTTCCTCAGTCCCAGCAT-3′; ULK1 (F) 5′-CAGACAGCCTGATGTGCAGT-3′ and (R) 5′-CAGGGTGGGGATGGAGAT-3′; GAPDH (F) 5′-GTCTCCTCTGACTTCAACAGCG-3′ and (R) 5′-ACCACCCTGTTGCTGTAGCCAA-3′. Three separate experiments were conducted on each sample. The standard curve method was used to analyze the data, and the relative mRNA expression of the gene of interest was normalized to GAPDH expression.

### 2.8. Immunocytochemical Staining

We cultured approximately 0.8–1 × 10^5^ Huh7 cells on 25 mm glass coverslips (Matsunami Glass Co., Osaka, Japan) in 35 mm dishes for 3–5 days. After four hours of BCAA treatment, the cells were washed with 1 × PBS and then fixed with 4% paraformaldehyde by incubation at room temperature for 20 min (min). Following three washes with 1 × PBS, fixed cells were incubated with a blocking solution (1% bovine serum albumin and 1% donkey serum in PBS) at room temperature for 30 min. Then they were incubated overnight with LC3 (rabbit) antibody at 4 °C. After being washed three times with 1 × PBS, the cells were incubated with Alexa Fluor 488 donkey anti-rabbit IgG (H + L) as a secondary antibody for 1 h at room temperature protected from light. For nuclear staining, 4′,6-Diamidino-2-phenylindole (DAPI) dihydrochloride (Dojindo, Kumamoto, Japan) was used. The cells were affixed in PermaFluor Mountant (Thermo Fisher Scientific). Stained images were obtained with a confocal microscope (LSM880; Carl Zeiss, Oberkochen, Germany) at a 20-fold magnification. The Zen software program (Carl Zeiss, Zen blue 3.2 edition) was used to process the images.

### 2.9. Cell Proliferation Assay

Flow cytometry (FACS) utilizing Ki67 labeling was used to assess the impact of BCAA on cell proliferation. In short, 1 × 10^5^ Huh7 cells were grown in a 35 mm culture dishes for 3 days. Four hours following BCAA treatment in both the DMEM and RPMI systems, cells were trypsinized and rinsed with 1 × PBS. The zombie aqua dye was used to stain the dead cells and eliminate them. Then, according to protocol B for the FoxP3 buffer solution (Thermo Fisher Scientific), the cells were fixed, permeabilized, and stained with an anti-Ki67 antibody. After washing twice, FACS buffer (1% heat-inactivated fetal calf serum, 1 mM EDTA, 0.05% NaN_3_) with 1% 7-AAD was added to the cells and incubated on ice for 5 min. Stained cells were examined using a flow cytometer (BD FACSVerse™; BD Biosciences, Becton Drive, Franklin Lakes, NJ, USA) and data were analyzed by FlowJo software (10.4.1 version) to determine the percentage of Ki67-positive cells.

### 2.10. Statistical Analyses

We used Student’s *t*-test for the statistical analyses, and the threshold of statistical significance was set at *p*  <  0.05. Unless otherwise stated, all experiments were performed three times. The data are shown as the mean  ±  standard deviation (SD).

## 3. Results

### 3.1. BCAA Upregulated PDCD4 Protein Levels

In this experiment, we used two medium systems, RPMI and DMEM without BCAA, which were supplemented with BCAA, as described in the Materials and Methods Section, to test the effect of the material. As shown in Figure 1, BCAA upregulated PDCD4 protein levels in both the RPMI (Figure 1a) and DMEM (Figure 1b) systems. It was shown that BCAA individual amino acids, especially leucine, also show similar pharmacological activities comparable to BCAA [4,44]. As shown in Figure 1a, individual amino acids leucine (Leu), isoleucine (Ile), and valine (Val) can also upregulate the protein levels (Figure 1a). However, the PDCD4 mRNA levels were downregulated by the BCAA treatment of Huh7 cells (Figure 1c). The results indicated that the PDCD4 protein expression levels are controlled pos-transcriptionally, potentially through the modulation of protein synthesis or through a proteasomal and/or autophagic degradation system.

### 3.2. BCAA Upregulated the Phosphorylation of mTORC1 Substrates and the Levels of Autophagy-Related Factors

BCAA treatment stimulated the phosphorylation of mTORC1 substrate 4E-BP1 (Figure 2a) and p70S6K1 (Figure 2b and Appendix A), stimulating cap-dependent translation and PDCD4 degradation, respectively. The phosphorylation of p70S6K1 substrate ribosomal protein S6 was also upregulated (Figure 2a) by BCAA. Rapamycin inhibited the phosphorylation of p70S6K1 upregulated by BCAA, as shown in Figure 2b–d. AKT-phosphorylation was not significantly increased by treatment with BCAA or individual amino acids in Huh7 cells (Appendix A). These results indicated that BCAA activates mTORC1 in Huh7 cells. p70S6K1 protein levels may be controlled by both proteasomal and autophagic degradation systems in Huh7 cells because both the proteasomal inhibitor MG132 and the autophagy inhibitor BafillomycinA1 upregulated protein levels (Figure 2b–d), indicating that the enzyme is degraded in both degradation systems. The inhibition of p70S6K1 phosphorylation by rapamycin (Figure 2b–d) indicated BCAA may induce the PDCD4 degradation via mTORC1 activation. Therefore, we investigated the mechanisms of whether mTORC1-mediated signaling contributes to PDCD4 degradation systems.

As shown in Figure 3a, in the absence of FBS from the medium, the protein levels of PDCD4 and the autophagy-related factors p62 and autophagy-related gene 5 (ATG5) were downregulated in the absence of BCAA compared to those in the cells cultured with complete medium containing FBS, but were higher in the presence of BCAA than in the absence of BCAA. In the presence of BCAA (×2, ×4), ATG5 was downregulated by rapamycin, indicating the stimulation of macroautophagy, but, in the absence of BCAA (×0), rapamycin showed no effect, for an unknown reason. MG132 upregulated PDCD4 levels in the presence of BCAA (×4), indicating that the proteasome system was activated. The levels of the autophagy-related proteins p62 and ATG5 were upregulated after inhibiting autophagy by Bafilomycin A1 (Figure 3a). In the autophagy system, except for canonical macroautophagy, ATG5-independent alternative autophagy, microautophagy, and chaperone-mediated autophagy (CMA) were working. The upregulation of p62 and ATG5 by BCAA was less effective than that by Bafilomycin A1, indicating that BCAA may not inhibit all kinds of autophagy. In the case of microtubule-associated protein 1A/1B-light chain 3 (LC3), which is also a marker of autophagic flux, on the withdrawal of FBS, the ratio of LC3-II to LC3-I increased in the absence of BCAA compared with a normal culture with FBS and decreased by BCAA (Figure 3b,c). The levels of LC3-II also decreased by BCAA, indicating that the formation of LC3-II from LC3-I is stimulated in the serum-free medium without BCAA and inhibited by the addition of BCAA (Figure 3b–d). Rapamycin treatment increased the LC3-II levels (Figure 3d). These results indicate that BCAA may inhibit autophagy in a rapamycin-sensitive manner.

### 3.3. ULK1knockdown Upregulated PDCD4 and the p62 Protein Levels

It is well known that autophagy initiation is controlled by the phosphorylation of unc-51-like kinase 1 (ULK1) and autophagosome formation is initiated by the activation of ULK1 [48]. ULK1 knockdown was performed using three ULK1-specific siRNAs in Huh7 cells and knockdown was confirmed by mRNA expression (Figure 4a). PDCD4 mRNA expression was not altered by ULK1 knockdown (Figure 4a). ULK1 knockdown cells were analyzed by Western blotting. The protein levels of PDCD4 and autophagy-related factor p62 were upregulated (Figure 4b), indicating that the degradation of PDCD4 protein may be inhibited by impaired autophagy in ULK1 knockdown Huh7 cells.

### 3.4. The Phosphorylation of ULK1 at S757 Was Stimulated by BCAA

It has been reported that the phosphorylation of ULK1 at S757 by mTORC1 inhibits autophagosome formation [49]. As shown in Figure 5a, the ULK1 protein (MW 150) levels were upregulated in the presence of the proteasome inhibitor MG132 or the autophagy inhibitor Bafilomycin A1, and downregulated in the presence of the mTORC1 inhibitor rapamycin. The protein amount of the MW 68 bands decreased in the cells with a higher protein amount of MW 150 bands in the presence of proteasome inhibitor MG132 or autophagy inhibitor Bafilomycin A1 and thus increased in the cells with a lower amount of MW 150 bands in the absence of inhibitor or in the presence of rapamycin (Figure 5a). The knockdown of ULK1 by specific siRNA treatment decreased the amount of the MW 68 band as well as that of the MW 150 band (Figure 5b). These results indicate that the MW 68 band may be a degradation product of MW 150 ULK1. Analyses of ULK1 phosphorylation at S757 revealed that the phosphorylation of MW 150 ULK1 was enhanced by BCAA and inhibited by rapamycin (Figure 5c,e). The phosphorylation of the MW 68 band was also upregulated by BCAA and downregulated by rapamycin (Figure 5c,f), indicating that the bands should also contain the structure around S757 of ULK1. These results indicate that BCAA by inhibiting the autophagy system induces the phosphorylation of ULK1 at S757 via the activation of mTORC1.

The PDCD4 protein levels were upregulated in the presence of the mTORC1 inhibitor, rapamycin, despite the expectation of autophagy activation (Figure 5c,d). This often happens and it may be due to the impaired activation of p70S6K1 by mTORC1 (Figure 2b), which is required for the phosphorylation of PDCD4 degron and the degradation of the protein in both autophagy and proteasome systems, as described previously [41,43].

It was previously reported that, during starvation, PDCD4 protein is degraded in a p62-dependent manner that mediates the selective autophagy in Huh7 cells and ATG5-independent-alternative macro-autophagy is involved in *ATG5*-deficient Huh7 cells [43]. We also observed that, in *ATG5*-deficient cells, LC3-I could not be converted into LC3-II; however, BCAA upregulated the protein levels of PDCD4 and LC3-I, similar to that in wild-type cells (Figure 6). In the presence of rapamycin, the PDCD4 protein levels increased in a similar way as in wild-type Huh7 cells. However, unlike the wild-type cells- where the LC3-I protein levels changed in response to rapamycin, the LC3-I levels in the *ATG5*-mutant cells remain unchanged (Figure 6). The phospho-ULK1 (S757) levels were also upregulated by BCAA and downregulated by rapamycin (Figure 6). The alterations in the protein levels by BCAA tested in the mutant cells were similar to those in wild-type Huh7 cells, except for the inhibition of the formation of LC3-II from LC3-I. The phosphorylation of ULK1 by mTORC1 may also inhibit autophagosome formation for ATG5-independent alternative autophagy [50,51,52,53].

### 3.5. BCAA Downregulated the Formation of LC3 Particles

To investigate the effect of BCAA on autophagic flux, LC3 particle formation was analyzed by immunocytochemistry. The withdrawal of FBS from the culture medium increased the number of LC3-particle-positive cells in the absence of BCAA and reduced the number of LC3-positive cells in the presence of BCAA in both DMEM and RPMI culture systems (Figure 7 and Appendix A). The number and intensity of the particles decreased in the cells cultured with BCAA compared to those cultured without BCAA (Figure 7a and Appendix A). These results suggest an inhibitory effect of BCAA in the autophagy system.

### 3.6. BCAA Inhibited Proliferation Activity

BCAA has been previously reported to inhibit the growth of HepG2 hepatoma cells [9]. In the present study, we assessed cell proliferation using a Ki67 marker. A FACS analysis revealed that the percentage of Ki67-positive cells increased under serum-free culture conditions without BCAA (×0). The increased cell proliferation in ×0 compared to controls (C) may be due to the downregulation of PDCD4 in ×0, as shown in Figure 1b. BCAAs partially inhibited cell proliferation in both DMEM (Figure 8) and RPMI (Appendix A) systems. This inhibition may be associated with increased PDCD4 protein levels (Figure 1b), consistent with previous reports showing that PDCD4 overexpression suppresses cell proliferation [54,55,56,57,58,59].

## 4. Discussion

Our findings showed that BCAA upregulates the PDCD4 protein levels by inhibiting the macroautophagy (autophagy) system via the activation of mTORC1 (Figure 9). Our data demonstrated that BCAA increased phosphorylation of mTORC1 substrates such as 4E-BP1 and p70S6K1 as well as the p70S6K1 substrate s6, indicating the activation of mTORC1. Further, BCAA upregulated the expression of autophagy-related factors p62 and ATG5 and downregulated the LC3-II formation from LC3-I. The results indicate BCAA inhibits the autophagy system. Our data show that PDCD4 protein levels were increased but the mRNA levels were not, indicating the protein expression was controlled post-transcriptionally such as by protein synthesis or degradation levels. PDCD4 protein is shown to be degraded in both the proteasome and autophagy systems [37]. Therefore, the results indicate PDCD4 protein may be upregulated by inhibiting the degradation in the autophagy system. Autophagy is initiated by the activation of ULK1. ULK1 forms a complex with ATG13-FIP200 and ATG101. AMPK1 phosphorylates and activates ULK1 in the complex, and activated ULK1 initiates the formation of an autophagosome [49,60]. Upon mTORC1 activation and the phosphorylation of ULK1 at S757 by mTORC1, the phosphorylation and activation of ULK1 by AMPK1 are impaired, thus inhibiting autophagosome formation and the autophagy system [49]. Our present data demonstrate that BCAA increases the phosphorylation of ULK1 at S757 in a rapamycin-sensitive manner, indicating that AMPK1 is not able to associate with ULK1 to phosphorylate the enzyme for activation because ULK1 is phosphorylated at S757 by mTORC1 (Figure 9). Thus, the initiation of autophagosomes is inhibited, and the autophagy system is impaired. As a result, the PDCD4 protein degradation in the autophagy system decreased and the protein levels were upregulated. Furthermore, our data show that ULK1 is degraded in both the proteasome and autophagy systems. Therefore, autophagy may be controlled not only by phosphorylation, but also by the abundance of ULK1.

The PDCD4 protein levels may be controlled by the proteasome and autophagy degradation systems because the protein levels were upregulated by both the proteasome inhibitor MG132 and the autophagy inhibitor Bafilomycin A1 [30,43]. The PDCD4 protein is phosphorylated at serine in the Degron sequence by p70S6K1 and PKC-delta or -epsilon, ubiquitinated, and degraded in the proteasome [41]. The ubiquitinated PDCD4 protein is navigated to autophagosomes by p62 and is then degraded in the autophagy system [43]. Our present data showed that the PDCD4 protein levels were often upregulated in the presence of rapamycin, despite the fact that autophagy should be activated under these conditions, and therefore the PDCD4 protein levels should be downregulated. This is because the p70S6K1 activation by mTORC1 was inhibited by rapamycin, impairing the phosphorylation of serine in the degron, which is necessary for the degradation of PDCD4 protein (Figure 9). Furthermore, p70S6K1 is also degraded in both the proteasome and autophagy systems; therefore, decreasing the enzyme amount may at least partly cause an increase in the PDCD4 levels in the presence of rapamycin. Altogether, the PDCD4 protein levels are controlled by the abundance and activity of enzymes involved in the degradation systems. Because both the overexpression and knockdown of PDCD4 induce cell death in Huh7 cells [38,61], the homeostasis of PDCD4 is important for cell survival. Our current data showed that, in the presence of MG132, BCAA upregulated the PDCD4 protein level, suggesting that the proteasome degradation system may be activated in the cells. These findings indicate that both the autophagy and proteasome degradation systems may be working in coordination. Therefore, the PDCD4 protein levels may be controlled by the many factors mentioned above, such as the activity of regulatory enzymes, in addition to environmental factors like serum, TPA, EGF, etc. [41,42].

In ATG5-deficient Huh7 cells, canonical macroautophagy is impaired, but the PDCD4 protein is degraded by ATG5-independent alternative autophagy [43]. PDCD4 protein levels were also upregulated by BCAA in a manner similar to that observed in wild-type cells. mTORC1 may also inhibit autophagosome formation in ATG5-independent alternative autophagy [50,51,52,53].

PDCD4 functions as a tumor suppressor [30], and PDCD4 knockdown mice develop spontaneous lymphomas and show a significantly reduced life span compared with wild-type siblings [40]. It was shown that BCAA supplements ameliorate cirrhosis tumors [11] and BCAA inhibits HepG2 hepatoma cell growth [9]. Takegoshi et al. reported that BCAA inhibits pro-fibrotic signaling and tumorigenesis by inhibiting TGF-β signaling, thereby preventing the development of HCC [62]. It is well reported that PDCD4 overexpression suppressed cell proliferation in breast carcinoma, hepatocellular carcinoma, nasopharyngeal carcinoma, glioma, pancreatic cancer, and mesenchymal stem cells [54,55,56,57,58,59]. Conversely, embryonic fibroblast (MEF) cells in PDCD4 knockout mice (Pdcd4 ^−^/^−^) proliferate faster than that of wild-type mice (Pdcd4 ^+^/^+^) [63]. Moreover, PDCD4 downregulation stimulates cell proliferation in colon carcinoma and glioblastoma-derived cells, thereby facilitating tumor growth [64,65]. Therefore, based on our FACS analysis, the upregulation of the PDCD4 protein levels by BCAA may make it useful as an anti-tumor supplement. However, PDCD4 knockdown mice are resistant to inflammatory diseases such as autoimmune encephalomyelitis and diabetes [40]. It has been reported that PDCD4 deficiency prevents diet-induced obesity, adipose tissue inflammation, and insulin resistance in PDCD4-deficient (PDCD4 ^−^/^−^) mice [66]. Another group also reported that PDCD4 deficiency ameliorated left ventricular remodeling and insulin resistance in a rat model of type 2 diabetic cardiomyopathy [67]. In polycystic ovary syndrome patients, a higher PDCD4 expression is associated with insulin resistance, lipid metabolism disorders, and granular cell apoptosis [68]. In obese individuals, the levels of circulating BCAA are elevated, and the elevated circulating BCAA levels are related to future insulin resistance [12,13], and reducing the BCAA intake ameliorates insulin resistance [14]. Therefore, PDCD4 may be associated with the induction of inflammatory diseases and insulin resistance and BCAA intake may exacerbate the inflammatory diseases via the upregulation of PDCD4.

## 5. Conclusions

BCAA inhibits the autophagy degradation system, impairing the formation of autophagosomes by phosphorylating the initiator of autophagosome ULK1 at serine 757 via the activation of mTORC1. The degradation of PDCD4 protein in the autophagy system was inhibited, and the protein levels were upregulated in Huh7 hepatoma cells. Our data showed that BCAA upregulates the tumor suppressor PDCD4 and reduces the proliferation marker Ki67. While BCAA intake may have potential benefits in cancer prevention and therapy, the upregulation of PDCD4 could pose a risk of worsening insulin resistance or diabetes. So, BCAA has to be carefully used for anti-tumor supplements, considering the context of patients. This concern thus requires further investigation in future studies.

## Figures and Tables

**Figure 1 cells-14-01975-f001:**
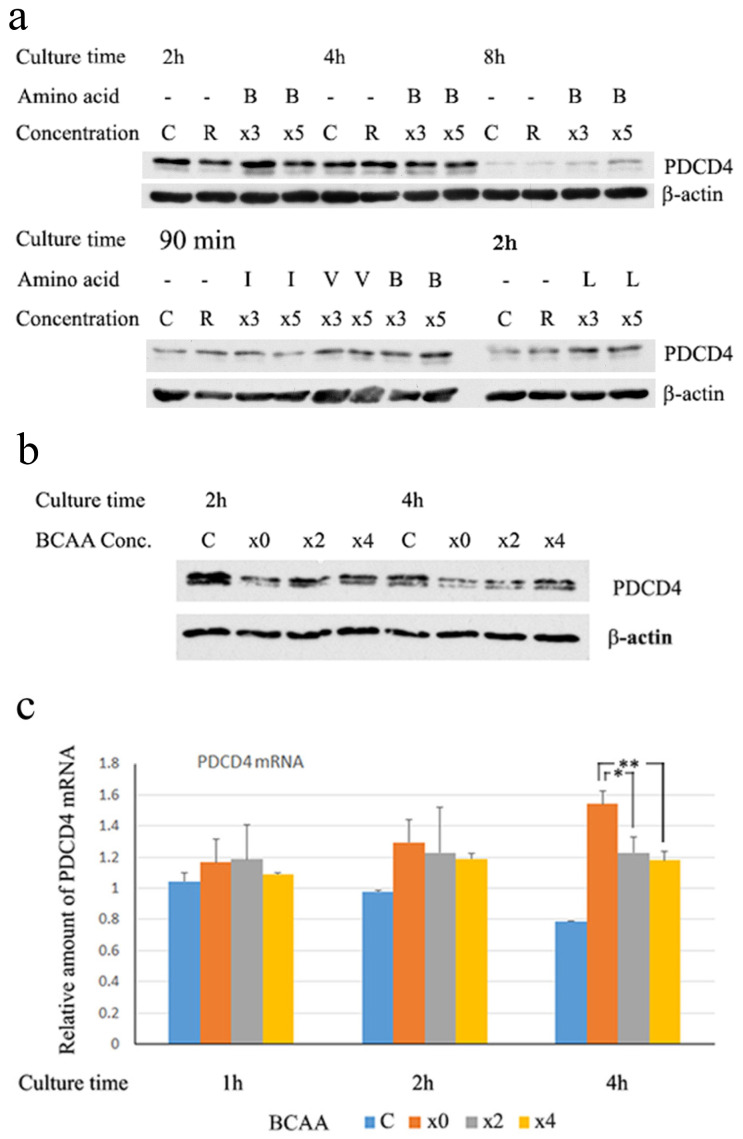
BCAA (B) upregulated the PDCD4 protein levels but downregulated the mRNA expression. (**a**) Huh7 cells were treated with the RPMI medium system containing different concentrations of BCAA [R(RPMI), ×3 and ×5] or the BCAA individual members leucine (L), isoleucine (I), and valine (V) together with C, cells cultured in DMEM containing FBS, for times indicated in the figure. The PDCD4 protein levels of the cells were then analyzed by Western blotting using the rabbit anti-PDCD4 antibody. All of the BCAA and the individual member amino acids upregulated PDCD4 protein levels. (**b**) Huh7 cells were treated with DMEM containing different concentrations of BCAA (×0, ×2 and ×4) together with C, cells cultured in DMEM containing FBS, and then analyzed by Western blotting using mouse anti-PDCD4 antibody. (**c**) Huh7 cells were treated with BCAA in the same way as (**b**) and PDCD4 mRNA levels were determined by a qRT-PCR analysis. Data are expressed as the mean ± SD obtained from three independent experiments. *t*-test: * *p* < 0.05, ** *p* < 0.01.

**Figure 2 cells-14-01975-f002:**
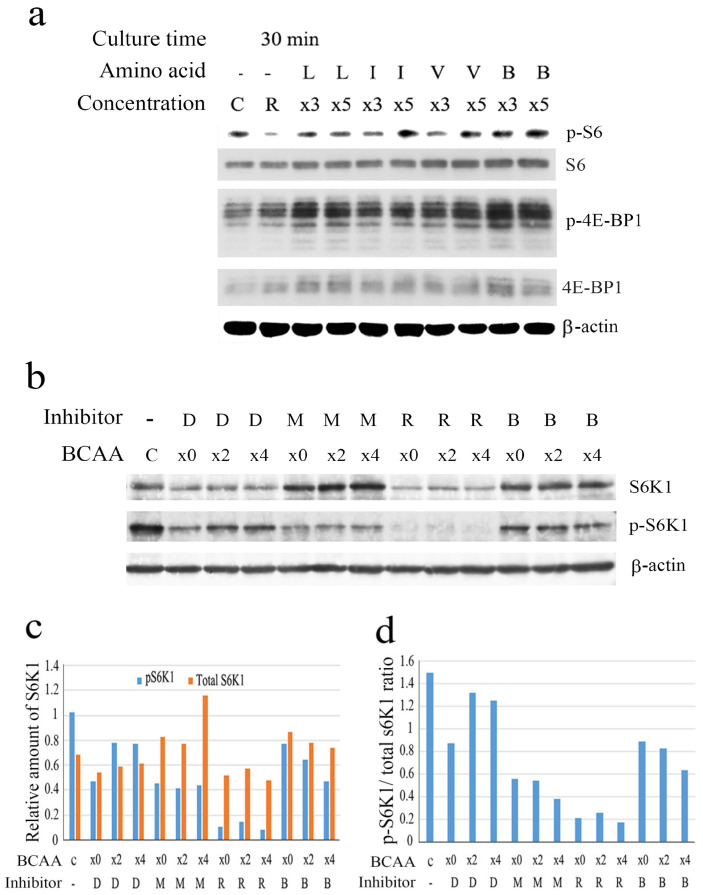
BCAA upregulated the phosphorylation of mTORC1 downstream substrates. (**a**) After culturing for 4 days, Huh7 cells were incubated for 30 min with the RPMI medium without FBS containing different amounts [R(RPMI), ×3 and ×5] of leucine (L), isoleucine (I), valine (V), and BCAA (B) together with C, cells cultured in DMEM containing FBS, and then were analyzed by Western blotting using anti-phospho-S6 (p-S6), total S6, anti-phospho-4E-BP1 (p-4E-BP1), total 4E-BP1, and anti-β-actin antibodies. (**b**) After a 4-day culture, Huh7 cells were further cultured for 4 h in the DMEM medium containing different amounts of BCAA (×0, ×2 and ×4) without FBS in the presence of DMSO (D), 20 μM MG132 (M), 0.2 nM rapamycin (R), or 10 nM Bafilomycin A1 (B) together with C, cells cultured in DMEM containing FBS, and the cells were then analyzed by Western blotting using anti-p70S6K1 (S6K1), anti-phospho-p70S6K1 (p-S6K1), and anti-β-actin antibodies. (**c**) Relative amounts of p70S6K1 (S6K1) and phospho-p70S6K1 (p-S6K1) obtained from (**b**). The amounts were normalized by β-actin. (**d**) Phospho-p70S6K1/total p70S6K1 (p-S6K1/S6K1) ratio calculated from data of (**c**). The phosphorylation of p70S6K1 was upregulated by BCAA and inhibited by rapamycin. Similar results were reproducibly obtained.

**Figure 3 cells-14-01975-f003:**
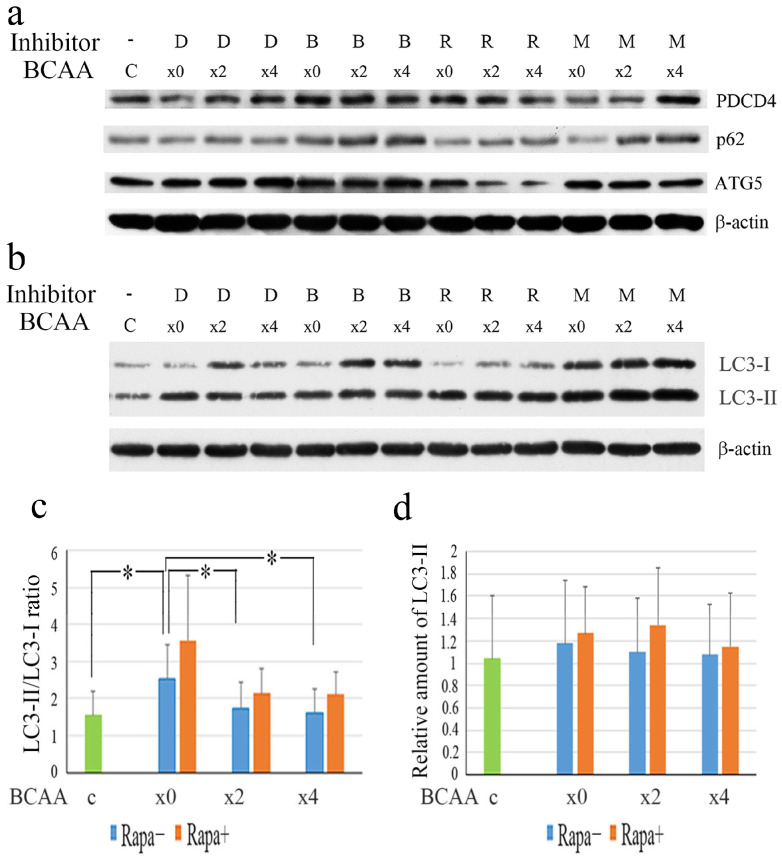
BCAA modulated the protein levels of the autophagy-related factors. (**a**) p62 and ATG5 protein levels were upregulated by BCAA. After a 4-day culture, Huh7 cells were treated for 4 h in serum-free DMEM containing different concentrations of BCAA (×0, ×2 and ×4) in the presence of DMSO (D), 10 nM Bafilomycin A1 (B), 0.2 nM rapamycin (R), or 20 μM MG132 (M) together with C, cells cultured in DMEM containing FBS. About 15–30 μg of protein sample was used per well and analyzed by Western blotting using anti-PDCD4, anti-p62, anti-ATG5, and anti-β-actin antibodies. The experiments were repeated 3 times and a representative result of them is shown in the figure. Similar results were obtained across all replicates. (**b**) LC3-II formation was decreased by BCAA. Huh7 cells treated the same way as (**a**) were analyzed by Western blotting using anti-LC3 and anti-β-actin antibodies. The experiments were repeated 3 times and a representative result of them is shown in the figure. (**c**) LC3-II/LC3-I ratio obtained from (**b**). (**d**) Relative amounts of LC3-II bands obtained from (**b**). Data of the mean ± SD obtained from three independent experiments are shown. *t*-test: * *p* < 0.05.

**Figure 4 cells-14-01975-f004:**
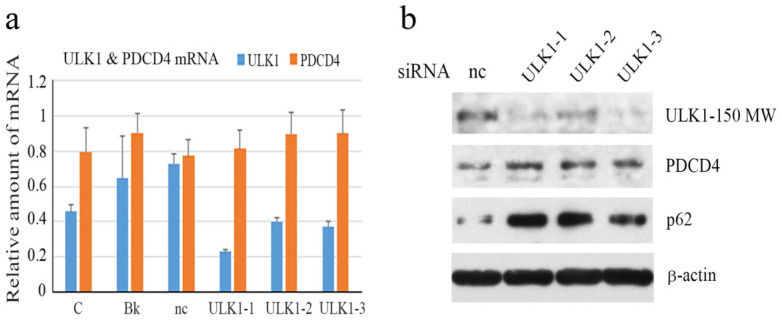
ULK1 knockdown upregulated PDCD4 and p62 protein levels. (**a**) Relative amounts of ULK1 and PDCD4 mRNA. Huh7 cells were treated with a negative control (nc), ULK1-1, ULK1-2, and ULK1-3 siRNAs and a medium containing the transfection reagents without siRNA (Bk) as a control. After culturing for 24 h, the cells were analyzed by qRT-PCR together with the cells cultured with normal medium (C). (**b**) ULK1 knockdown upregulated PDCD4 and p62 protein levels. Huh7 cells were treated with a negative control (nc), ULK1-1, ULK1-2, and ULK1-3 siRNAs and cultured for 24 h. The cells were then subjected to a Western blotting analysis using anti-ULK1, anti-PDCD4, anti-p62, and anti-β-actin antibodies.

**Figure 5 cells-14-01975-f005:**
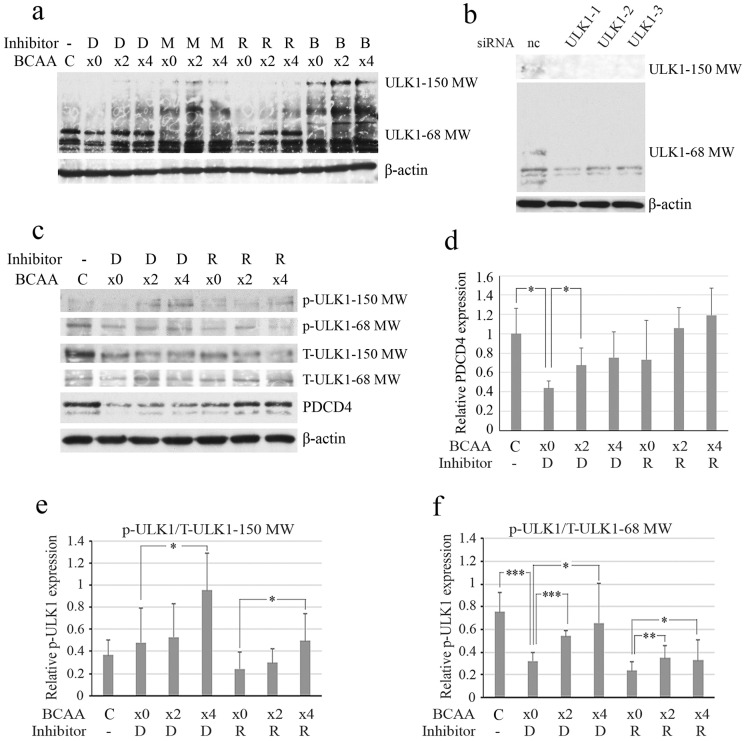
BCAA increased the phosphorylation of ULK1 at S757 and rapamycin decreased the phosphorylation. (**a**) After a 4-day culture, Huh7 cells were treated for 4 h with serum-free DMEM containing different concentrations of BCAA (×0, ×2 and ×4) in the presence of DMSO (D), 20 μΜ MG132 (M), 0.2 nM rapamycin (R), or 10 nM Bafilomycin A1 (B) together with C, cells cultured in DMEM containing FBS. The cells were then analyzed by Western blotting using anti-ULK1 and anti-β-actin antibodies. ULK1 protein (MW 150) bands were upregulated by both the proteasome inhibitor MG132 and the autophagy inhibitor bafilomycin A_1_. (**b**) ULK1 knockdown. After a 4-day culture, Huh7 cells were treated with negative control (nc), ULK1-specific ULK1-1, ULK1-2, and ULK1-3 siRNAs. The cells were then subjected to immunoblotting using anti-ULK1 and anti-β-actin antibodies. Both MW 150 bands and MW 65 bands were downregulated by the ULK1 knockdown. (**c**) Huh7 cells were treated the same as (**a**) with different concentrations (×0, ×2 and ×4) of BCAA in the presence of DMSO (D) or 0.2 nM rapamycin (R) together with C, cells cultured in DMEM containing FBS, and analyzed by Western blotting using anti-ULK1 (T-ULK1), anti-phospho-ULK1 (S757) (p-ULK1), and anti-β-actin antibodies. The phosphorylation of both MW 150 and 68 bands were upregulated by BCAA. The experiments were repeated 3 times and a representative result of them is shown in the figure. (**d**) Relative amounts of PDCD4 obtained from (**c**). (**e**) The ratio of p-ULK1/T-ULK1 of MW 150 obtained from (**c**). (**f**) The ratio of p-ULK1/T-ULK1 of MW 68 obtained from (**c**). Data of the mean ± SD obtained from three independent experiments are shown. *t*-test: * *p* < 0.05, ** *p* < 0.01, *** *p* < 0.001.

**Figure 6 cells-14-01975-f006:**
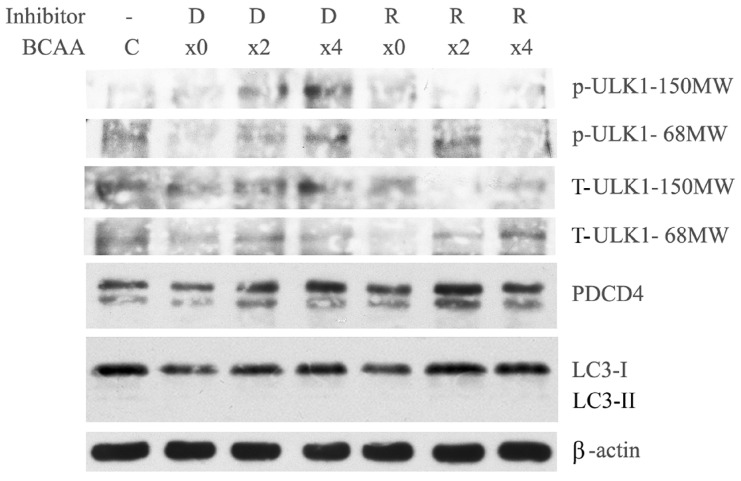
BCAA induced the mTORC1-mediated phosphorylation of ULK1 at S757 and upregulated the PDCD4 protein levels in *ATG5*-deficient Huh7 cells. *ATG5*-deficient Huh7 cells were cultured for 4 days, followed by a 4 h treatment with serum-free DMEM containing different concentrations of BCAA (×0, ×2, ×4) in the presence of DMSO (D) or 0.2 nM rapamycin (R) together with C, cells cultured in DMEM containing FBS. Western blotting was performed using the antibodies shown in the figure. Phospho-ULK1 (S757) levels at both 150 and 68 MW were elevated by BCAA and were reduced by rapamycin. Conversion of LC3-I to LC3-II was inhibited; however, BCAA treatment increased the protein levels of PDCD4 and LC3-I. This figure represents the data from two reproducible individual experiments.

**Figure 7 cells-14-01975-f007:**
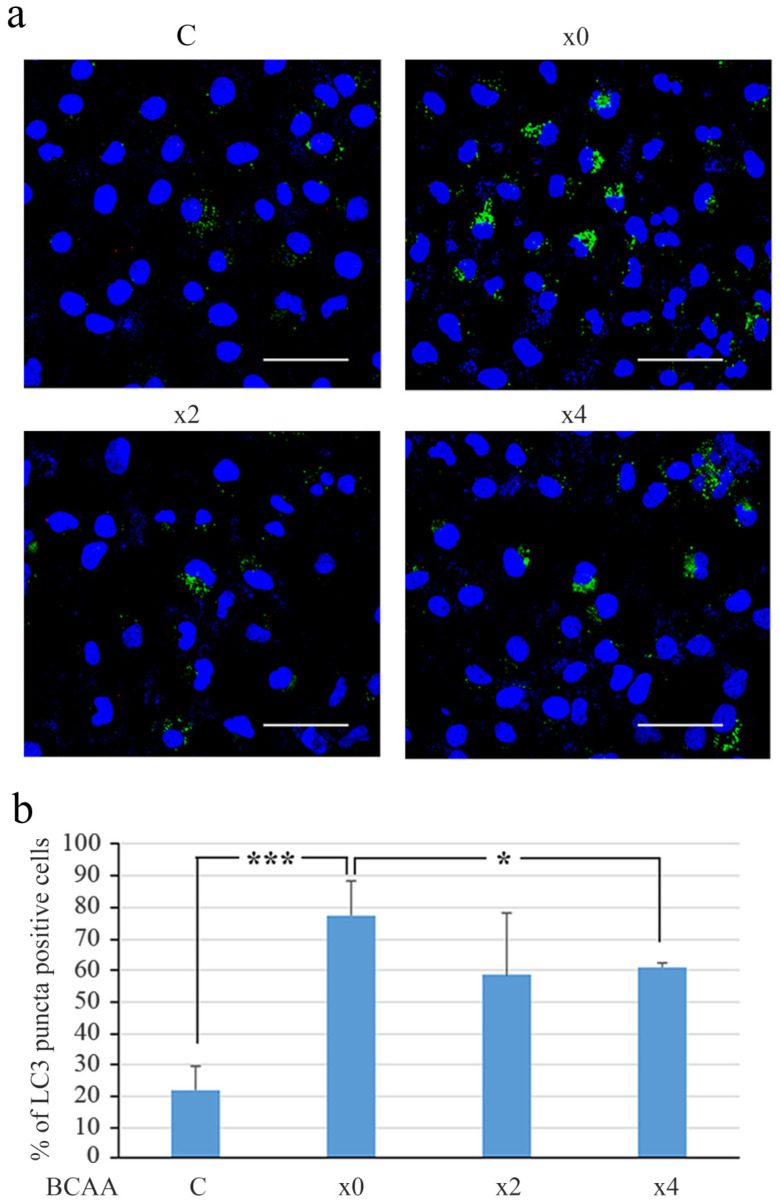
The formation of LC3 particles was inhibited by the BCAA treatment of Huh7 cells in the DMEM medium system. Immunocytochemical staining of LC3. Huh7 cells were treated with serum-free DMEM containing various concentrations of BCAA (×0, ×2, ×4) together with cells cultured in normal DMEM medium (C) and stained with rabbit anti-LC3 antibody, as described in the Materials and Methods Section. (**a**) Immunocytochemical staining of LC3 (green) with nuclei counterstained in blue. To quantify LC3-particle-positive cells, 12 fields, each containing 80–150 cells, were photographed in different areas. Scale bar indicates 100 µm. (**b**) The percentage of LC3-particle-positive cells was calculated, and the results are presented as the mean ± SD across 12 fields. *t*-test: * *p* < 0.05, *** *p* < 0.001.

**Figure 8 cells-14-01975-f008:**
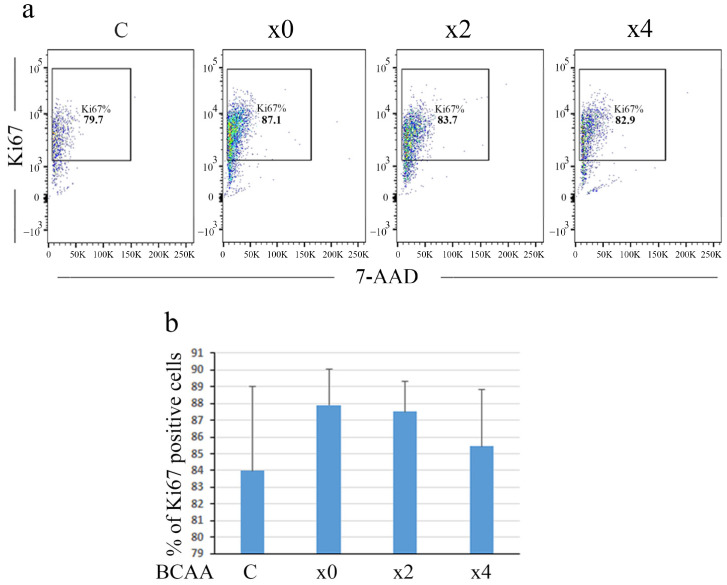
BCAA partially inhibited cell proliferation of Huh7 cells in DMEM systems. FACS analysis of BCAA-treated Huh7 cells. After a 3-day culture period, Huh7 cells were treated with serum-free DMEM containing different concentrations of BCAA (×0, ×2 and ×4) for 4 h. As a control, the cells were also treated with FBS-containing medium (C). Following treatment, cells were stained with anti-Ki67 antibody and 7-AAD, as described in Section 2, and subsequently analyzed by flow cytometry. After gating to remove dead and doublet cells, singlet cells were analyzed based on Ki67 and 7-AAD staining. (**a**) Gating strategy of one representative experiment out of three repeated experiments. Each dot represents Ki67-positive single cell. (**b**) The average percentage of Ki67-positive cells in three separate experiments.

**Figure 9 cells-14-01975-f009:**
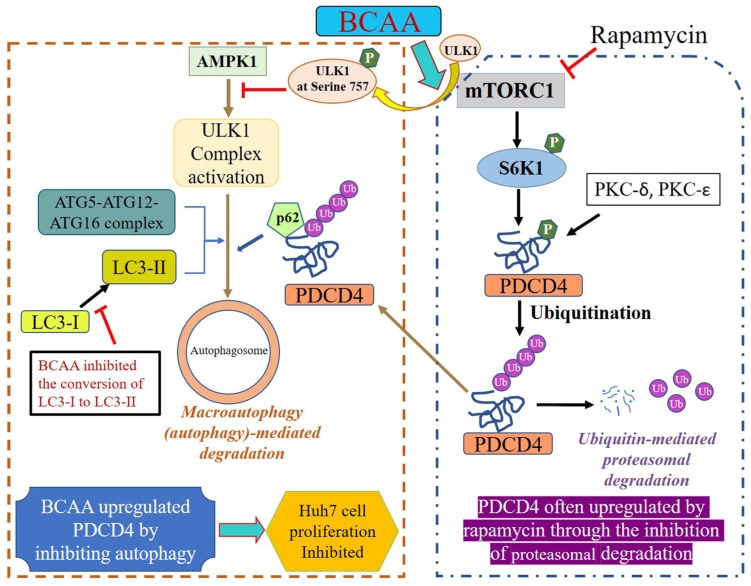
Schematic diagram of molecular mechanism of PDCD4 degradation in the presence of BCAA in Huh7 cells. BCAA stimulates ULK1 phosphorylation at serine 757 via the activation of mTORC1. This in turn inhibits the activation of ULK1 complex and the conversion of LC3-I to LC3-II, thereby inhibiting autophagosome formation and consequently autophagy-mediated degradation of PDCD4. In the presence of rapamycin, PDCD4 is often upregulated through the inhibition of p70 S6K1 (S6K1)-mediated phosphorylation and ubiquitination of PDCD4, which leads to inhibition of PDCD4 degradation through the proteasomal system. This results in reduced Huh7 cell proliferation.

## Data Availability

The original contributions presented in this study are included in the article/Appendix A. Further inquiries can be directed to the corresponding authors.

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
