# Peer review of "BCAA (Branched-Chain Amino Acids) Inhibiting the Autophagy System via the Activation of mTORC1, Thereby Upregulating the Tumor Suppressor PDCD4 in Huh7 Hepatoma Cells"

_cells, 2025, doi:10.3390/cells14241975_

Round 1

Reviewer 1 Report

Comments and Suggestions for Authors

This manuscript reports that branched-chain amino acids (BCAAs) upregulate the tumor suppressor PDCD4 by inhibiting autophagy through mTORC1-mediated ULK1 phosphorylation. The findings may offer new mechanistic insights into how BCAAs regulate PDCD4 expression. However, several points in the manuscript require clarification and improvement.

  1. The figures and/or figure labels of 2C, 2D, 3C, 3D, 4A, 5, and 8 are too small to read.
  2. What are the medium conditions for “C” and “R” in Figures 1 and 2? From my understanding, “C” may refer to culture medium without FBS; however, the meaning of “R” is unclear. The authors should define all abbreviations in each figure legend.
  3. The authors should discuss why PDCD4 protein levels decrease from C to X0 in Figure 1B, and why PDCD4 protein levels decrease while mRNA levels increase in Figure 1C.
  4. How many replicates were performed for the Western blots in Figures 3A and 3B? The band intensities of p62 and ATG5 show only slight differences between C and D-X0.
  5. In Figure 5C, PDCD4 protein levels do not appear to increase upon BCAA treatment. Quantification of band intensities may be necessary to support this result.
  6. It is surprising that cell proliferation (Ki67-positive) in the presence of FBS (C) is lower than in the absence of FBS (X0) in figure 8. Is this difference statistically significant?
  7. A summary diagram illustrating the major findings or pathways would be helpful.

Reviewer 2 Report

Comments and Suggestions for Authors

The authors investigated branched-chain amino acids (BCAA) and effects on PDCD4 tumor suppressor in Huh7 hepatoma cells. Fig. 1 shows a small increase in PDCD4 protein with BCAA treatment. Fig. 2 investigates effects of BCAA on mTOR1 pathway S6K and 4E-BP1. Fig. 3 investigates effects of BCAA on autophagy pathway p62, ATG5, and LC3. Figs. 4-6 investigates ULK1 (initial member of autophagy pathway). Fig. 7 shows BCAA increase in autophagosomes and Fig. 8 shows BCAA effect on proliferation.

In general, the paper provides useful data linking BCAA with PDCD4 and autophagy/mTOR pathways. A better overall connection of these topics, summarized by a schematic, would be helpful in putting things together, based on the data presented. Otherwise, the paper is difficult to follow, and the overall message difficult to find. Some questions, comments, and suggestions are intended to improve the paper.

  1. Introduction: Provide a better rationale for investigating PDCD-4. Better link to BCAA and mTOR pathway.
  2. Fig. 1: Clarify that B stands for BCAA., R for RPMI, C control ? Fig. 1C y-axis should be protein not mRNA. Otherwise, PDCD-4 mRNA goes up, not down.
  3. Fig. 2C, D should be larger, easier for reader to see. A better explanation or significance of this result.
  4. Fig. 3: Interpretation of data is complicated. Without B, R, or M, BCAA addition does not have much effect on autophagy markers p62 or ATG5. Only with artificial addition of B, R, or M is BCAA effect observed. Please comment.
  5. A schematic or figure describing the results of this paper would be helpful. Otherwise, as presented, it is difficult to follow.
  6. Discussion needs improvements in "discussing" results and how they may be interpreted. Or why results are significant in relation to clinical application.
  7. Can also mention BCAA supplements that are easily available and perhaps should be used with caution.
Comments on the Quality of English Language

English writing is OK.

Reviewer 3 Report

Comments and Suggestions for Authors

Perveen and colleagues propose that BCAAs activate mTORC1 in Huh7 hepatoma cells, leading to ULK1 S757 phosphorylation, inhibition of autophagy, and stabilization of the tumor suppressor PDCD4. They also report reduced Ki67 positivity as evidence of antiproliferative effects. Mechanistic connections are supported by rapamycin sensitivity, ULK1 siRNA, ATG5-deficient cells, and changes in p62/LC3 readouts.

Although the manuscript is promising, the authors should address the following questions before it can be considered for publication:

  • Why are both RPMI and DMEM media used with the same HuH7 cells?
  • In Fig. 1A, BCAA treatment is shown at two concentrations (X3 and X5). However, PDCD4 levels do not appear higher in X5 compared to X3; both look similar to control. If there is no incremental effect, why did the authors use two concentrations (X3 and X5)?
  • In Fig. 1B, PDCD4 protein appears decreased compared to control, which seems inconsistent with the reported results. Could the authors clarify?
  • Why are different time intervals used across figures—2, 4, and 8 h in Fig. 1A, but only 2 and 4 h in Fig. 1B? Please make the time intervals consistent. Also, is there any published literature supporting specific time intervals or concentrations of BCAA, leucine (L), isoleucine (I), and valine (V)?
  • PDCD4 mRNA levels are decreased while PDCD4 protein levels are increased after BCAA addition. Why are the mRNA and protein results inconsistent?
  • Please include total protein levels of S6 and 4E-BP1 in Fig. 2A to complement the phosphorylated forms.
  • Why did the authors measure both total BCAA and the individual amino acids (L, I, V)? Please clarify the rationale.
  • Please enlarge Figs. 2C and 2D so that readers can clearly see the data.

Author Response

" Please see the attachment"

Round 2

Reviewer 1 Report

Comments and Suggestions for Authors

No further comments

Reviewer 2 Report

Comments and Suggestions for Authors

The authors have addressed concerns appropriately.

Comments on the Quality of English Language

Addition of new text requires editing. 

Reviewer 3 Report

Comments and Suggestions for Authors

The author has satisfactorily addressed all my previous comments and concerns. 
